# Analysis of the Origin of Emiratis as Inferred from a Family Study Based on *HLA-A*, *-C*, *-B*, -*DRB1*, and *-DQB1* Genes

**DOI:** 10.3390/genes14061159

**Published:** 2023-05-26

**Authors:** Zain Al Yafei, Abdelhafidh Hajjej, Marion Alvares, Ayeda Al Mahri, Amre Nasr, Rajaa Mirghani, Ali Al Obaidli, Mohamed Al Seiari, Steven J. Mack, Medhat Askar, Hisham A. Edinur, Wassim Y. Almawi, Gehad ElGhazali

**Affiliations:** 1Sheikh Khalifa Medical City-Union71-Purehealth, Abu Dhabi P.O. Box 51900, United Arab Emirates; zalyafei@union.ae (Z.A.Y.); malvares@union71.ae (M.A.); ayeda@union71.ae (A.A.M.); 2United Arab Emirates University, Al Ain P.O. Box 51900, United Arab Emirates; 3Department of Immunogenetics, National Blood Transfusion Center, Tunis P.O. Box 1006, Tunisia; 4College of Medicine, King Saud Bin Abdulaziz University for Health Sciences, Riyadh P.O. Box 22490, Saudi Arabia; nasra@ksau-hs.edu.sa; 5Higher College of Technology, Abu Dhabi P.O. Box 25026, United Arab Emirates; 6SEHA Kidney Care, SEHA, Abu Dhabi P.O. Box 92900, United Arab Emirates; 7Department of Pediatrics, University of California, San Francisco, Oakland, CA 94609, USA; 8Qatar University, Doha P.O. Box 2713, Qatar; 9School of Health Sciences, Universiti Sains Malaysia, Health Campus, Kubang Kerian 16150, Kelantan, Malaysia; edinur@usm.my; 10Faculty of Sciences, El-Manar University, Tunis P.O. Box 94, Tunisia; wassim.almawi@outlook.com

**Keywords:** Arabs, Emiratis, genotyping, human leukocyte antigen, HLA, allele, haplotypes

## Abstract

In this study, we investigated HLA class I and class II allele and haplotype frequencies in Emiratis and compared them to those of Asian, Mediterranean, and Sub-Saharan African populations. Methods: Two-hundred unrelated Emirati parents of patients selected for bone marrow transplantation were genotyped for HLA class I (*A*, *B*, *C*) and class II (*DRB1*, *DQB1*) genes using reverse sequence specific oligonucleotide bead-based multiplexing. HLA haplotypes were assigned with certainty by segregation (pedigree) analysis, and haplotype frequencies were obtained by direct counting. HLA class I and class II frequencies in Emiratis were compared to data from other populations using standard genetic distances (SGD), Neighbor-Joining (NJ) phylogenetic dendrograms, and correspondence analysis. Results: The studied HLA loci were in Hardy–Weinberg Equilibrium. We identified 17 *HLA-A*, 28 *HLA-B*, 14 *HLA-C*, 13 *HLA-DRB1*, and 5 *HLA-DQB1* alleles, of which *HLA-A*02* (22.2%), -*B*51* (19.5%), -*C*07* (20.0%), -*DRB1*03* (22.2%), and -*DQB1*02* (32.8%) were the most frequent allele lineages. *DRB1*03*~*DQB1*02* (21.2%), *DRB1*16*~*DQB1*05 (17.3%)*, *B*35*~*C*04* (11.7%), *B*08*~*DRB1*03* (9.7%), *A*02*~*B*51* (7.5%), and *A*26*~*C*07*~*B*08*~*DRB1*03*~*DQB1*02* (4.2%) were the most frequent two- and five-locus HLA haplotypes. Correspondence analysis and dendrograms showed that Emiratis were clustered with the Arabian Peninsula populations (Saudis, Omanis and Kuwaitis), West Mediterranean populations (North Africans, Iberians) and Pakistanis, but were distant from East Mediterranean (Turks, Albanians, Greek), Levantine (Syrians, Palestinians, Lebanese), Iranian, Iraqi Kurdish, and Sub-Saharan populations. Conclusions: Emiratis were closely related to Arabian Peninsula populations, West Mediterranean populations and Pakistanis. However, the contribution of East Mediterranean, Levantine Arab, Iranian, and Sub-Saharan populations to the Emiratis’ gene pool appears to be minor.

## 1. Introduction

The human leukocyte antigen (HLA) complex is a group of over 200 closely linked genes, located on the short arm of human chromosome 6, spanning 3.6 Mb [1]. HLA genes are extremely polymorphic, with over thirty-six thousand alleles at multiple loci described [1]. HLA genes encode for the cell-surface HLA molecules involved in recognizing processed peptide antigens presented to T lymphocytes [2]. There is a strong linkage disequilibrium (LD) between alleles at different HLA loci, under which specific alleles are inherited together more frequently than expected. HLA allele and haplotype frequencies vary greatly among different populations and ethnic groups [3,4], which makes comparison of HLA frequencies between populations valuable for anthropological studies and understanding the history of human migrations.

The United Arab Emirates (UAE) is situated at the South-East corner of the Arabian Peninsula, west of the Strait of Hormuz, bordering Saudi Arabia to the west and south, and Oman to the east (Figure 1). The UAE spreads over 83,600 km^2^ area, includes 200 islands, and consists of 7 Emirates which entered into federation in 1971. The current UAE population is about 9.60 million (2018) [5], 11.48% of whom are indigenous Emiratis [6]. Arabic is the official language of the country and is spoken by the indigenous population.

Artifacts discovered at Jebel Faya in the Emirate of Sharjah show that the territory has been inhabited for the past 125,000 years. Neolithic village settlement and Late Stone Age artefacts were also discovered in the Marawah and Baynunah Islands [7]. Historically, the UAE is a central link for connecting Africa with Asia, and the first human migration out of Africa to Asia passed across the UAE [8,9]. More recent migrations into the Arab Peninsula from the Middle East and Asia took place across the Straits of Hormuz, a major trade channel linking the Indian subcontinent and the Gulf States. This migration out of Africa may have had a bearing on the genetic makeup of the present-day Emirati population. The Arab migration in 7th and 11th centuries contributed slightly to the homogenization of the Arabian Peninsula (Saudis, Kuwaitis, Emiratis) and African populations [10]. The notion of the pre-Islamic relatedness between North Africans and Peninsular Arabs is favored by the documented relatedness of Emiratis with Berbers and Basques who remained isolated and did not undergo external genetic exchanges during the 7th and 11th centuries’ Arab migration [10]. It may also be attributed to the fact that Peninsular Arabs are distant from the Arab Levantines, although the Arab migration were earlier, massive, frequent and effective in the Levant [10].

This paper aims to investigate the genetic relatedness between Emiratis and other populations. This is the largest study that examines the origin of present-day Emiratis using HLA family data. We hypothesize that Emiratis are more closely related to Arabian Peninsula, West Mediterranean and Indian subcontinent populations than to East Mediterranean, Levantine Arab, Iranian, and Sub-Saharan populations.

## 2. Material and Methods

### 2.1. Study Subjects

Study subjects included 200 unrelated healthy parents from 100 indigenous UAE families, with the average number of siblings in families ranging from 2 to 7. Only families with four haplotypes well-defined by segregation were included in this study. All grandparents of the study subjects lived in the UAE. For comparative purposes, populations from Arabian Peninsula, Asia, Africa and Europe were included, and are detailed in the Appendix A. The study received an ethical approval from the Institutional Review Board (IRB) at Sheikh Khalifa Medical City in Abu Dhabi.

### 2.2. DNA Genotyping

Total genomic DNA was prepared from anti-coagulated peripheral venous blood using the QIAmp Blood Mini kit (Qiagen, Hilden, Germany) or the MagNa Pure 96 DNA and Viral NA small volume kit (Roche Diagnostics, Mannheim, Germany) according to the manufacturers’ protocol. DNA samples were stored below −20 °C until analysis. The DNA concentration was quantified using a NanoDrop 2000 C spectrophotometer (ThermoFisher Scientific, Wilmington, DE, USA). HLA class I (*A*, *C* and *B*) and class II (*DRB1* and *DQB1*) genotyping was performed at low-to-intermediate resolution using Luminex LabType Sequence Specific Probe Hybridization (SSO) typing kits (OneLambda Inc., Thermo Fisher, Canoga Park, CA, USA) on a Luminex 200 or Luminex Flexmap 3D instrument, following the manufacturer’s instruction. Briefly, the target DNA was PCR-amplified by the reverse single specific oligonucleotide using group-specific biotinylated primers specifically designed for exons 2 and 3 of HLA class I (*HLA*-*A*, -*C* and -*B*) and exon 2 for HLA class II (*HLA-DRB1* and -*DQB1*) genes. Biotinylated PCR product was denatured, hybridized to probes in beads and detected by R-Phycoerythrin-conjugated Streptavidin (SAPE) in the Luminex flow analyzer. The instrument has a red laser for beads identification and a green one for phycoerythrin detection. Positive and negative controls were included in each set of beads in order to subtract non-specific background signals and normalize the raw data for possible sample variability and reaction efficiency.

To ensure accuracy of the results, quality control measurements comprised of re-typing 10% of samples with the most common haplotypes using the One Lambda NXType HLA sequencing assay and TypeStream Virtual NGS analysis software v3.0 (One Lambda, Inc., Canoga Park, CA, USA) on Ion Torrent Sequencing System (Thermo Fisher Scientific, Waltham, MA, USA). Our HLA laboratory is accredited by the College of American Pathologists (CAP), American Society of Hostocompatibility and Immunogenetics (ASHI) and ISO15189.

### 2.3. Statistical Analysis

HLA haplotypes were assigned with certainty by segregation (pedigree) and haplotypes frequencies were obtained by direct counting. The *D*′ [11], *Wn* [12], *Wa*/*b* and *Wb*/*a* [13] measurements of the linkage disequilibrium (LD) were calculated for one-field (allele lineage level) haplotypes using the Phased Or Unphased Linkage Disequilibrium (POULD) R package (v1.0.1) (https://cran.r-project.org/web/packages/pould/pould.pdf), (accessed on 23 November 2022) [14,15]. The conditional asymmetric LD (ALD) measurements, *Wa*/*b* and *Wb*/*a*, are extensions of the *Wn* measurement for highly polymorphic loci. Where a *Wn* value may over-estimate the LD between such loci, the ALD approach allows the investigation of LD for pairs of polymorphic loci in which alleles at one locus may display complete LD with alleles at a second locus, while alleles at the second locus are in a less-than-complete LD with alleles at the first locus. The *Wa*/*b* measurement reflects a variation on alleles at the *a* locus on any of the haplotypes conditioned on *b* locus alleles, while the *Wb*/*a* measurement reflects variation on alleles at the *b* locus on any of the haplotypes conditions on *a* locus alleles. HLA allele frequency counts and Hardy–Weinberg Equilibrium (HWE) testing were done using PyPop (Python for Population) genomics, version 0.7.0 (http://www.pypop.org), (accessed on 21 June 2021) [16,17]. For Ewens–Watterson homozygosity (EWH), PyPop was also used in calculating observed (*F*_obs_) and expected (*F*_exp_) homozygosity, and the normalized deviate of the homozygosity (*F*_nd_) for each locus [18,19,20,21]. DISPAN software was used to construct phylogenetic trees (dendrograms) [22] using the neighbor-joining (NJ) method [23,24,25] from matrices of standard genetic distances (SGDs) [25]. Three-dimensional correspondence analysis and bi-dimensional representation based on SGDs were performed using the VISTA version 7.2.8 software [26].

## 3. Results

### 3.1. HLA Allele Lineage Frequencies

The studied HLA-*A*, -*C*, -*B*, -*DRB1* and -*DQB*1 frequencies were in Hardy–Weinberg Equilibrium (HWE) in the UAE population (Appendix A). Frequencies of the allelic lineages of the *HLA-A*, *-C*, *-B*, *-DRB1* and *-DQB1* are shown in Table 1. Seventeen *HLA-A*, 28 *HLA-B*, 14 *HLA-C*, 13 *HLA-DRB1* and 5 *HLA-DQB1* allelic lineages were identified in the Emirati population. *HLA-A*02* (22.2%), *B*51* (19.5%) and *C*07* (22.0%) were the most frequent HLA class I allelic lineages in the Emirati families. For the HLA class II, *DRB1*03* (22.2%), *DQB1*02* (32.7%) and *DQB1*05* (29.7%) were the most frequent allelic lineages in the Emirati families.

### 3.2. Two- and Five-Locus HLA Haplotypes

Linkage Disequilibrium (LD) estimates for two-locus phased haplotypes are shown in Table 2. While *D*′ and *Wn* values are often comparable (c.f., values for A~DQB1 and C~DQB1 haplotypes), the ALD values illustrate asymmetry in the LD for specific locus pairs. The ALD measurements (*WDRB1*/*DQB1* and *WDQB1*/*DRB*) dissect the variation on each locus conditioned on the other. The highest ALD values are observed between DRB1 and DQB1 and between HLA-B and HLA-C; *WDRB1*~*DQB1* is 0.9, while *WDQB1*~*DRB1* is 0.58, and *WB*~*C* is 0.78, while *WC*~*B* is 0.64. The lowest LD values are observed between HLA-C and DQB1 and between HLA-A and DQB1; *WC*~*DQB1* is 0.31, while *WDQB1*~*C* is 0.19, and *WA*~*DQB1* is 0.22, while *WDQB1*~*A* is 0.13. In most cases, where there are large differences in the number of alleles at two loci, the less polymorphic locus displays a lower ALD when conditioned on the more polymorphic locus. Overall, the LD and ALD values are higher between more proximal loci (c.f., B~C vs. A~C, and A~DQB1 vs. DRB1~DQB1), reflecting higher recombination rates over longer physical distances [10].

In the Emirati families, *DRB1*03*~*DQB1*02* (21.3%), *DRB1*16*~*DQB1*05* (17.3%), *B*35*~*C*04* (11.7%), *B*08*~*DRB1*03* (9.7%) and *A*02*~*B*51* (7.5%) were the most frequent two locus haplotypes. Those with significant linkage disequilibrium and frequencies exceeding 1% are shown in Table 3 and the complete list of HLA-class I and -class II two-locus haplotypes are shown in the Appendix A.

*HLA- A*26*~*C*07*~*B*08*~*DRB1*03*~*DQB1*02* was the most frequent 5-locus HLA haplotype (4.25%) in the studied Emirati sample (Table 4).

To verify the HLA typing done, eighteen samples containing the most common extended haplotypes were re-typed by next generation sequencing (NGS); the haplotypes are listed below:

*A*26:01:01*~*B*08:01:01*~*C*07:02:01*~*DRB1*03:01:01*~*DQB1*02:01:01*, 

*A*02:05:01*~*B*50:01:01*~*C*06:02:01*~*DRB1*07:01:01*~*DQB1*02:02:01*, 

*A*03:01:01*~*B*52:01:01*~*C*12:02:02*~*DRB1*15:02:01*~*DQB1*06:01:01*,

*A*24:02:01*~*B*08:01:01*~*C*07:02:01*~*DRB1*03:01:01*~*DQB1*02:01:01*, 

*A*02:01:01*~*B*51:01:01*~*C*15:13:01*~*DRB1*04:02:01*~*DQB1*03:02:01*, 

*A*11:01:01*~*B*40:06:01*~*C*15:02:01*~*DRB1*16:02:01*~*DQB1*05:02:01*

*A*33:01:01*~*B*14:02:01*~*C*08:01:01*~*DRB1*01:01:01*~*DQB1*05:01:01.*

High resolution typing by NGS confirmed the accuracy of the low-to-intermediate resolution Luminex-based HLA typing.

### 3.3. Ewens–Watterson Homozygosity Test of Neutrality

Ewens–Watterson homozygosity (EWH) test results for Emiratis revealed negative *F*_nd_ values for the five loci tested, and lower than expected homozygosity under selective neutrality (Appendix A). Except for the *HLA-B* locus (*p* = 0.15), significant deviations were observed for *A* (*p* = 0.006), *C* (*p* = 0.003), *DRB1* (*p* = 0.005), and *DQB1* (*p* = 0.006). The Ewens–Watterson homozygosity test of neutrality findings suggest that the HLA allele frequency distributions were shaped by balancing selection, conforming with previous works carried out on a worldwide sample of populations [27].

### 3.4. Phylogenetic and Correspondence Analysis

To construct Neighbor-Joining (NJ) dendrograms and correspondence plots, standard genetic distance (SGD) values are used. SGD values between Emiratis and other populations are shown in Table 5. Omani (0.0276), Saudi (0.0886), North African [as Tunisian (0.1108) and Algerian (0.1127)] and Iberian populations had the closest genetic distance from the investigated Emirati population, while East Mediterranean and Sub-Saharan populations had the highest genetic distance.

NJ dendrograms, based on SGD using *HLA-DRB1* allele frequency data from 85 population datasets (Table 5), demonstrated relatedness between Emiratis and other populations (Figure 2). The NJ tree shows clustering of the Emirati population with the Arabian Peninsula (Omanis, and Kuwaitis) and West Mediterranean (North Africans) populations. However, they appear distant from Turkish, Macedonian, Greek, Levantine (Syrians, Palestinians, and Lebanese), Iranian, Iraqi Kurdish, and African Sub-Saharan populations.

Correspondence analysis based on SGD, presenting the relationship between Emiratis and several populations from the Arabian Peninsula, Asia, Africa and Europe according to *HLA-A*, -*B*, -*DRB1*, and -*DQB1* allele frequency data is shown in Figure 3. Similar trends were demonstrated, with clustering of the 35 populations into 3 distinct groups. Emiratis cluster with the Arabian Peninsula, Pakistani, North African and Iberian populations.

## 4. Discussion

This work constitutes an anthropological genetic study on Emiratis and is unique in that it involves HLA family data, a large sample size relative to the total indigenous population, and genotypes for five HLA loci. This was in contrast to earlier studies of the UAE population, which often had small sample sizes, and some even relied on serological HLA data [28,29,30,31]. HLA allelic frequencies of the present study were compared with approximately 90 populations from the three continents.

*HLA-A*02* (22.2%) was the most frequent *HLA-A* allelic lineage in Emiratis. *A*02* was also frequent in other Arab populations—Saudis (30.4%), Tunisian Berbers (29.3%), Moroccans (26.2%), Sudanese (25.9%), Iraqi Arabs (14.77), Jordanians (21.3%), and Omanis (21.6%) [32]. This lineage was also observed at comparable frequencies in Mediterranean populations, including Spanish (23.0%), French (Rennes region; 27.0%), and Italians (25.4%) [32]. *HLA- B*51* (19.5%) was the most frequent *HLA-B* allelic lineage in Emiratis and was observed at comparable frequencies in Saudis (19.3%), Iraqi Kurds (15.6%), Turks (15.8%), Sudanese (21.8%), Tunisian (12.2%), Albanian (17.2%), and Greek (15.4%) populations [32]. It is worth mentioning that *HLA-B51* is strongly and consistently associated with an increased risk for Behçet’s Disease [33].

In addition, *HLA-C*07* was the most common *HLA-C* allelic lineage in Emiratis (22.0%), and in Saudi (24.9%), Iranian (19.6%), Iraqi Kurd (18.9%), French (25.8%), Italian (24.12%), and Spanish (Murcia region; 25.8%) populations [32]. For class II alleles, *DRB1*03* (22.2%) was the most frequent *DRB1* allelic lineage in Emiratis. *DRB1*03* was previously reported with similar frequencies in Saudi (15.8%), Moroccan (17.1%), Sardinian (27.2%), Iraqi Arab (14.43%), and Turkish (13.0%) populations [28]. Furthermore, *HLA-DQB1*02* (32.7%) and -*DQB1*05* (29.7%) were the most frequent *DQB1* allelic lineage in Emiratis. *DQB1*02* was also observed in Iraqi Arab (30.9%), Saudi (35.2%), Moroccan (29.7%), French (26.9%), Spanish (30.8%), and Romanian (27.6%) populations [32].

The most frequent two-locus haplotypes in Emiratis were also seen in Arab and North African populations. *HLA-A*02*~*B*51* (7.5% in Emiratis) was also detected in Pakistani (6.2%), Moroccan (1.14%), Iranian (3.9%), and Macedonian (5.42%) populations [28]. In addition, *B*35*~*C*04* (11.7%), a common haplotype in Emiratis, was also frequent in Pakistani (13.3%) and Macedonian (14.0%) populations [32]. Similarly, *B*08-DRB1*03* (9.7% in Emiratis) was also reported for Moroccan (2.1%), Italian (2.9%), and Macedonian (5.4%) populations, and *DRB1*03*~*DQB1*02*—a frequent two-locus HLA haplotype (21.2%)—was also seen in Moroccan (9.0%) and Pakistani (21%) populations [32]. The *HLA-DRB1*03-DQB1*02* haplotype was shown to confer susceptibility to type 1 Diabetes [34].

*HLA-A*26*~*C*07*~*B*08*~*DRB1*03*~*DQB1*02* was the most frequent 5-locus HLA haplotype (4.25%) in the studied Emirati sample and was also detected in Iraqi Kurdish (2.4%), Iranian (7.8%), and Indian (5.9%) populations [32]. It was the second most common five- locus haplotype in the Kuwaiti population, while *A*02*~*C*06*~*B*50*~*DRB1*07*~*DQB1*02* was the most common haplotype [35,36]. These haplotype results are concordant with those obtained by allele frequency analysis. It is noteworthy that 18 samples containing the most common extended haplotypes were re-typed by high-resolution and included the *HLA-A*26:01:01*~*C*07:02:01*~*B*08:01:01*~*DRB1*03:01:01*~*DQB1*02:01:01* haplotype. A recent small-sized study was concordant with our current findings [31]. This haplotype ranked third and fourth in the U.S., South Asian Indian, and South East Asian populations, respectively (National Marrow Donor Program [NMDP] [37]. It is also the second most common five-locus haplotype in the Kuwaiti population [35]. It has been previously suggested that the *HLA-A1* allele in the *A1∼C7∼B8∼ DR17∼DQ2* (most frequent in Europeans) was replaced by A26 in the Emirati population [38].

The distribution of *HLA*-*A*, -*C*, -*B*, -*DRB1*, and -*DQB1* genotypes in Emiratis was compared with those of other Arab, Mediterranean, and Sub-Saharan African populations using genetic distances, NJ dendrograms, and correspondence analysis (Figure 2 and Figure 3). Our results showed that Emiratis appear to be related to West Mediterranean (North African, Iberian, and French) and Arabian Peninsula (Saudi, Kuwaiti, Omani) populations, and relatively distinct from Levantines, Mediterranean East Europeans, Iranian, and Sub-Saharan communities. This suggests a low genetic contribution of Levantine Arabs, Iranians, and Sub-Saharans to the Emirati gene pool.

The relatedness of Emiratis to neighboring Peninsular Arab communities is explained by the fact that Arab Peninsula countries share, with slight differences, similar historical background and the same territory. On the other hand, the relatedness of Emiratis to North African and Iberian populations is likely the result of the mass eastern migration which took place 10,000 years earlier, after the settlement of hyper-arid conditions in the African Sahara [39].

## 5. Conclusions

Our study, based on genetic distance, NJ dendrograms, and correspondence analysis, showed that Emiratis are related to the Arabian Peninsula, Pakistani, and West Mediterranean populations, but distinct from Levantine, Iranian, and Sub-Saharan communities. An important part of the Emirati genome came from the West.

## Figures and Tables

**Figure 1 genes-14-01159-f001:**
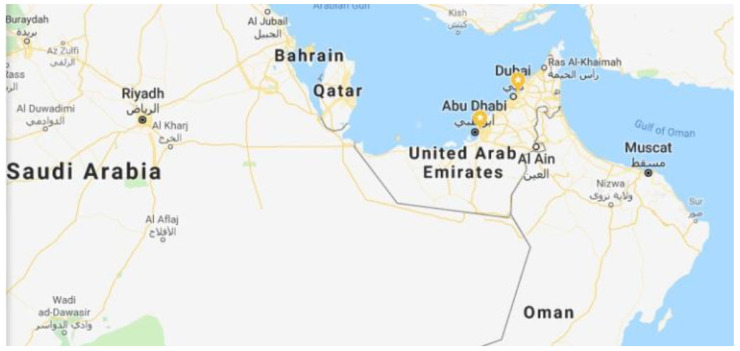
Map of the United Arab Emirates (UAE) and neighboring countries in the Arab Peninsula.

**Figure 2 genes-14-01159-f002:**
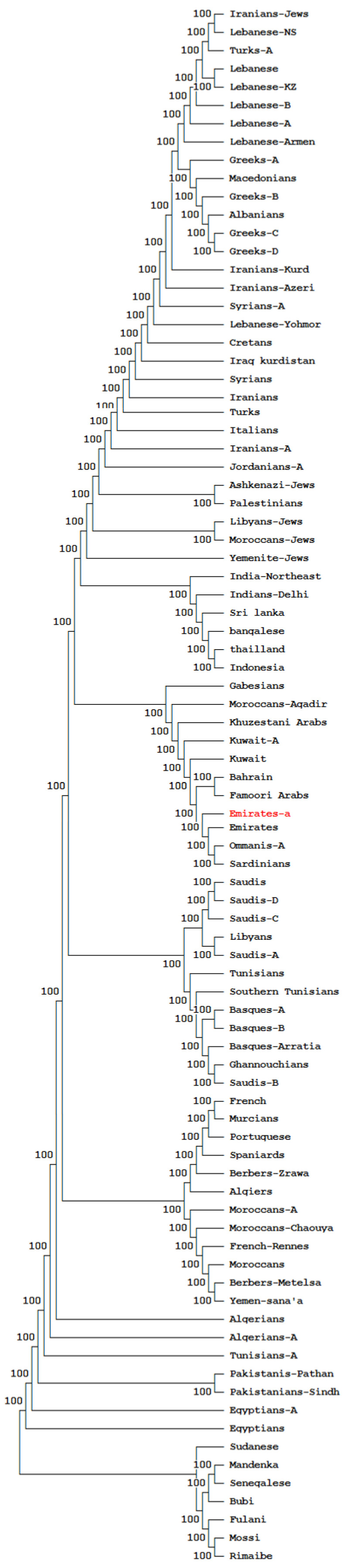
Neighbor-Joining dendrogram, based on Standard genetic distances (SGD), showing relatedness between Emiratis and other populations using HLA-DRB1 allele frequency data. Population data were taken from references detailed in the Appendix A. Bootstrap values from 1.000 replicates are shown. The studied population (Emiratis) is marked in red.

**Figure 3 genes-14-01159-f003:**
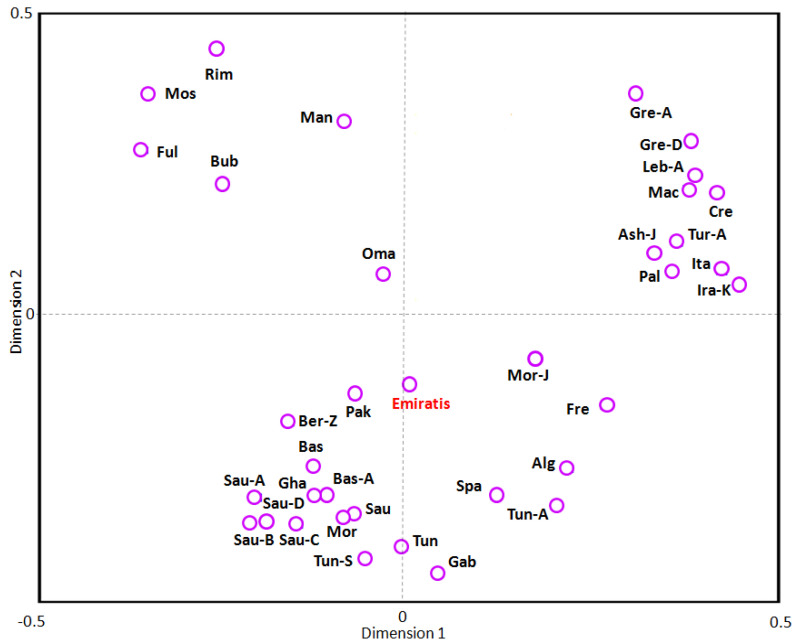
Correspondence analysis (bi-dimensional representation), based on the standard genetic distances, showing the relationship between Emiratis and worldwide populations according to HLA-A, -B, DRB1, and DQB1 allele frequencies data. Population data were taken from references detailed in the Appendix A. The studied population (Emiratis) is marked in red.

**Table 1 genes-14-01159-t001:** *HLA-A*, *-C*, *-B*, *-DRB1* and *-DQB1* Allele Frequencies in Emiratis (2n: 400).

*A*-Locus			*B*-Locus			*C*-Locus			*DRB1*-Locus			*DQB1*-Locus		
Allele	Frequency	SD *	Allele	Frequency	SD*	Allele	Frequency	SD *	Allele	Frequency	SD *	Allele	Frequency	SD *
A*02	0.2225	0.0208	B*51	0.1950	0.0198	C*07	0.2000	0.0200	DRB1*03	0.2225	0.0208	DQB1*02	0.3275	0.0235
A*11	0.0950	0.0147	B*35	0.1375	0.0172	C*04	0.1550	0.0181	DRB1*16	0.1775	0.0191	DQB1*05	0.2975	0.0229
A*26	0.0900	0.0143	B*08	0.1150	0.0160	C*06	0.1375	0.0172	DRB1*07	0.1275	0.0167	DQB1*03	0.2025	0.0201
A*01	0.0850	0.0140	B*50	0.0825	0.0138	C*15	0.1300	0.0168	DRB1*04	0.1025	0.0152	DQB1*06	0.1400	0.0174
A*03	0.0800	0.0136	B*58	0.0575	0.0117	C*12	0.0900	0.0143	DRB1*15	0.0975	0.0149	DQB1*04	0.0325	0.0089
A*24	0.0775	0.0134	B*40	0.0500	0.0109	C*03	0.0675	0.0126	DRB1*11	0.0700	0.0128			
A*32	0.0750	0.0132	B*14	0.0450	0.0104	C*16	0.0675	0.0126	DRB1*13	0.0550	0.0114			
A*33	0.0675	0.0126	B*52	0.0350	0.0092	C*08	0.0450	0.0104	DRB1*01	0.0450	0.0104			
A*68	0.0675	0.0126	B*18	0.0325	0.0089	C*14	0.0300	0.0085	DRB1*10	0.0375	0.0095			
A*30	0.0550	0.0114	B*15	0.0300	0.0085	C*02	0.0300	0.0085	DRB1*08	0.0225	0.0074			
A*31	0.0300	0.0085	B*41	0.0250	0.0078	C*17	0.0275	0.0082	DRB1*14	0.0225	0.0074			
A*23	0.0250	0.0078	B*53	0.0225	0.0074	C*01	0.0125	0.0056	DRB1*12	0.0125	0.0056			
A*74	0.0100	0.0050	B*37	0.0200	0.0070	C*05	0.0050	0.0035	DRB1*09	0.0075	0.0043			
A*66	0.0075	0.0043	B*73	0.0175	0.0066	C*18	0.0025	0.0025						
A*34	0.0050	0.0035	B*57	0.0175	0.0066									
A*29	0.0050	0.0035	B*07	0.0175	0.0066									
A*69	0.0025	0.0025	B*49	0.0175	0.0066									
			B*44	0.0150	0.0061									
			B*38	0.0125	0.0056									
			B*42	0.0125	0.0056									
			B*27	0.0100	0.0050									
			B*45	0.0100	0.0050									
			B*13	0.0075	0.0043									
			B*39	0.0050	0.0035									
			B*56	0.0025	0.0025									
			B*47	0.0025	0.0025									
			B*81	0.0025	0.0025									
			B*55	0.0025	0.0025									

* SD: Standard Deviation.

**Table 2 genes-14-01159-t002:** Linkage Disequilibrium estimates for two-locus phased haplotypes.

*Loc1~Loc2*	*D’*	*Wn*	*W_Loc1/Loc2_*	*W_Loc2/Loc1_*	N_Haplotypes
*A~B*	0.480	0.373	0.323	0.397	140
*A~C*	0.414	0.312	0.319	0.303	112
*A~DRB1*	0.309	0.235	0.239	0.234	117
*A~DQB1*	0.208	0.224	0.220	0.128	64
*B~C*	0.845	0.749	0.783	0.637	65
*B~DRB1*	0.5215	0.418	0.482	0.337	124
*B~DQB1*	0.432	0.469	0.469	0.219	84
*C~DRB1*	0.406	0.280	0.327	0.317	96
*C~DQB1*	0.308	0.319	0.305	0.193	54
*DRB1~DQB1*	0.907	0.820	0.895	0.583	26

**Table 3 genes-14-01159-t003:** Frequent (≥1%) HLA Class I and Class II 2-Locus haplotypes in Emiratis.

Haplotype	Frequency	SD *	D’	χ^2^	*p*-Value	Haplotype	Frequency	SD *	D’	χ^2^	*p*-Value
*A*~*B*						*B*~*C*					
*A*02*~*B*51*	0.0750	0.0132	0.21	14.72	1.0 × 10^−4^	*B*41*~*C*17*	0.0150	0.0061	0.59	125.70	<1.0 × 10^−5^
*A*26*~*B*08*	0.0525	0.0112	0.53	85.26	1.0 × 10^−5^	*B*53*~*C*04*	0.0150	0.0061	0.61	18.40	1.8 × 10^−5^
*A*33*~*B*58*	0.0250	0.0078	0.39	52.30	<1.0 × 10^−5^	*B*07*~*C*07*	0.0150	0.0061	0.82	19.23	1.2 × 10^−5^
*A*03*~*B*50*	0.0250	0.0078	0.25	24.31	<1.0 × 10^−5^	*B*57*~*C*06*	0.0125	0.0056	0.67	19.99	<1.0 × 10^−5^
*A*11*~*B*40*	0.0250	0.0078	0.45	40.16	<1.0 × 10^−5^	*B*38*~*C*12*	0.0125	0.0056	1.00	51.20	<1.0 × 10^−5^
*A*32*~*B*35*	0.0225	0.0074	0.19	7.22	7.2 × 10^−3^	*B*42*~*C*17*	0.0125	0.0056	1.00	179.06	<1.0 × 10^−5^
*A*24*~*B*08*	0.0200	0.0070	0.16	6.76	9.3 × 10^−3^	*B*15*~*C*02*	0.0100	0.0050	0.31	39.12	<1.0 × 10^−5^
*A*33*~*B*14*	0.0200	0.0070	0.40	42.55	<1.0 × 10^−5^	*B*45*~*C*16*	0.0100	0.0050	1.00	55.82	<1.0 × 10^−5^
*A*01*~*B*37*	0.0150	0.0061	0.73	46.42	<1.0 × 10^−5^	** *B* ** **~*DRB1***					
*A*03*~*B*52*	0.0150	0.0061	0.38	23.95	<1.0 × 10^−5^	*B*08*~*DRB1*03*	0.0975	0.0148	0.80	117.49	<1.0 × 10^−5^
*A*03*~*B*18*	0.0100	0.0050	0.25	9.46	2.1 × 10^−3^	*B*50*~*DRB1*07*	0.0475	0.0106	0.51	64.97	<1.0 × 10^−5^
*A*11*~*B*41*	0.0100	0.0050	0.34	11.10	9.0 × 10^−4^	*B*52*~*DRB1*15*	0.0300	0.0085	0.84	95.14	<1.0 × 10^−5^
** *B* ** **~** ** *C* **						*B*40*~*DRB1*16*	0.0275	0.0082	0.45	20.01	1.0 × 10^−5^
*B*35*~*C*04*	0.1175	0.0161	0.83	238.26	<1.0 × 10^−5^	*B*58*~*DRB1*03*	0.0250	0.0078	0.27	6.36	0.0117
*B*50*~*C*06*	0.0800	0.0136	0.96	210.04	<1.0 × 10^−5^	*B*14*~*DRB1*01*	0.0150	0.0061	0.30	36.43	<1.0 × 10^−5^
*B*51*~*C*15*	0.0700	0.0128	0.43	44.92	<1.0 × 10^−5^	*B*14*~*DRB1*07*	0.0150	0.0061	0.24	7.18	7.4 × 10^−3^
*B*51*~*C*16*	0.0525	0.0112	0.72	62.65	<1.0 × 10^−5^	*B*18*~*DRB1*16*	0.0150	0.0061	0.34	7.42	6.4 × 10^−3^
*B*14*~*C*08*	0.0450	0.0104	1.00	400.00	<1.0 × 10^−5^	*B*37*~*DRB1*10*	0.0125	0.0056	0.61	78.06	<1.0 × 10^−5^
*B*58*~*C*03*	0.0450	0.0104	0.77	198.26	<1.0 × 10^−5^	*B*41*~*DRB1*08*	0.0100	0.0050	0.43	66.45	<1.0 × 10^−5^
*B*40*~*C*15*	0.0400	0.0098	0.77	83.56	<1.0 × 10^−5^	*B*53*~*DRB1*04*	0.0100	0.0050	0.38	11.70	6.0 × 10^−4^
*B*52*~*C*12*	0.0325	0.0089	0.92	124.57	<1.0 × 10^−5^	*B*49*~*DRB1*13*	0.0100	0.0050	0.55	36.56	<1.0 × 10^−5^
*B*51*~*C*14*	0.0300	0.0085	1.00	51.07	<1.0 × 10^−5^	*B*15*~*DRB1*11*	0.0100	0.0050	0.28	13.18	3.0 × 10^−4^
*B*37*~*C*06*	0.0200	0.0070	1.00	51.21	<1.0 × 10^−5^	*B*42*~*DRB1*03*	0.0100	0.0050	0.74	9.76	1.8 × 10^−3^
*B*73*~*C*15*	0.0175	0.0066	1.00	47.68	<1.0 × 10^−5^	** *DRB1* ** **~*DQB1***					
*B*49*~*C*07*	0.0175	0.0066	1.00	28.50	<1.0 × 10^−5^	*DRB1*03*~*DQB1*02*	0.2125	0.0205	0.93	204.69	<1.0 × 10^−5^
*B*18*~*C*12*	0.0175	0.0066	0.49	33.00	<1.0 × 10^−5^	*DRB1*16*~*DQB1*05*	0.1725	0.0189	0.96	187.82	<1.0 × 10^−5^
*B*07*~*C*07*	0.0150	0.0061	0.95	185.92	<1.0 × 10^−5^	*DRB1*07*~*DQB1*02*	0.1100	0.0157	0.79	76.03	<1.0 × 10^−5^
*DRB1*04*~*DQB1*03*	0.0875	0.0141	0.82	119.94	<1.0 × 10^−5^	*DRB1*10*~*DQB1*05*	0.0375	0.0095	1.00	36.80	<1.0 × 10^−5^
*DRB1*15*~*DQB1*06*	0.0850	0.0140	0.85	192.21	<1.0 × 10^−5^	*DRB1*14*~*DQB1*05*	0.0225	0.0074	1.00	21.74	<1.0 × 10^−5^
*DRB1*11*~*DQB1*03*	0.0650	0.0123	0.91	98.28	<1.0 × 10^−5^	*DRB1*08*~*DQB1*04*	0.0125	0.0056	0.54	80.11	<1.0 × 10^−5^
*DRB1*13*~*DQB1*06*	0.0475	0.0106	0.84	101.25	<1.0 × 10^−5^	*DRB1*04*~*DQB1*04*	0.0100	0.0050	0.22	6.15	0.0131
*DRB1*01*~*DQB1*05*	0.0450	0.0104	1.00	44.51	<1.0 × 10^−5^	*DRB1*12*~*DQB1*03*	0.0100	0.0050	0.75	11.19	0.0008

* SD: Standard Deviation.

**Table 4 genes-14-01159-t004:** Most Frequent (≥0.75%) HLA 5-locus Haplotypes in Emiratis.

*A~B~C~DRB1~DQB1 Haplotype*	Frequency	Standard Deviation	*A~B~C~DRB1~DQB1 Haplotype*	Frequency	Standard Deviation
*A*26~B*08~C*07~DRB1*03~DQB1*02*	0.0425	0.0101	*A*11~B*51~C*07~DRB1*16~DQB1*05*	0.0100	0.0050
*A*02~B*50~C*06~DRB1*07~DQB1*02*	0.0150	0.0061	*A*32~B*08~C*07~DRB1*03~DQB1*02*	0.0100	0.0050
*A*03~B*52~C*12~DRB1*15~DQB1*06*	0.0150	0.0061	*A*01~B*37~C*06~DRB1*10~DQB1*05*	0.0100	0.0050
*A*02~B*08~C*07~DRB1*03~DQB1*02*	0.0150	0.0061	*A*11~B*41~C*07~DRB1*08~DQB1*03*	0.0100	0.0050
*A*24~B*08~C*07~DRB1*03~DQB1*02*	0.0150	0.0061	*A*03~B*50~C*06~DRB1*07~DQB1*02*	0.0075	0.0043
*A*01~B*51~C*15~DRB1*03~DQB1*02*	0.0125	0.0056	*A*68~B*35~C*04~DRB1*04~DQB1*03*	0.0075	0.0043
*A*33~B*14~C*08~DRB1*01~DQB1*05*	0.0125	0.0056	*A*03~B*50~C*06~DRB1*04~DQB1*04*	0.0075	0.0043
*A*02~B*51~C*15~DRB1*04~DQB1*03*	0.0125	0.0056	*A*01~B*58~C*03~DRB1*03~DQB1*02*	0.0075	0.0043
*A*11~B*40~C*15~DRB1*16~DQB1*05*	0.0125	0.0056	*A*26~B*51~C*14~DRB1*15~DQB1*06*	0.0075	0.0043
*A*33~B*58~C*03~DRB1*03~DQB1*02*	0.0100	0.0050	*A*11~B*40~C*15~DRB1*03~DQB1*02*	0.0075	0.0043
*A*68 B*14 C*08 DRB1*07 DQB1*02*	0.0100	0.0050	*A*32~B*18~C*12~DRB1*16~DQB1*05*	0.0075	0.0043
*A*33 B*58 C*03 DRB1*16 DQB1*05*	0.0100	0.0050	*A*02~B*51~C*16~DRB1*16~DQB1*05*	0.0075	0.0043
*A*02 B*35 C*04 DRB1*16 DQB1*05*	0.0100	0.0050			

**Table 5 genes-14-01159-t005:** Standard genetic distances (SGD) between Emiratis and other populations.

*HLA-A*, *-B*, *-DRB1*, and -*DQB1* (35 Populations)		*HLA-DRB1* (85 Populations)			
Population	SGD *	Population	SGD	Population	SGD
Oman	0.0276	Emirates	0.0012	Italy	0.2688
Saudi Arabia-B	0.0886	Kuwait-A	0.0456	Spain-Basques-B	0.2703
Saudi Arabia-C	0.0930	Oman-A	0.0589	Greece-Cretans	0.2738
Saudi Arabia	0.0982	Bahrain	0.0594	Iran-A	0.2799
Tunisia-A	0.1108	Sardinia	0.0622	Iraq-kurdistan	0.2899
Algeria	0.1127	Khuzestani	0.1068	Jordan-A	0.3086
Spain	0.1139	Algeria-A	0.1178	Egypt-A	0.3235
Saudi Arabia-D	0.1140	Kuwait	0.1186	India-Northeast	0.3269
Tunisia-Gabes	0.1246	Tunisia-A	0.1267	Iran-Kurd	0.3316
Morocco	0.1258	Morocco-Agadir	0.1310	Palestine	0.3322
France	0.1344	Algeria	0.1447	Greece-D	0.3470
Pakistan	0.1380	Algeria	0.1473	Egyptians	0.3498
Macedonia	0.1409	Saudi Arabia-C	0.1522	Greece-C	0.3568
Greece-Cretans	0.1436	Saudi Arabia-B	0.1530	Ashkenazi-Jews	0.3582
Italy	0.1442	Gabesians	0.1590	Syria-A	0.3704
Tunisia	0.1530	Tunisia	0.1643	Syria	0.3799
South Tunisia	0.1569	Saudi Arabia-D	0.1716	Albania	0.3813
Iraq-Kurdis	0.1598	Morocco-A	0.1747	Sudan	0.3844
Saudi Arabia-A	0.1732	Spain-Spaniards	0.1770	Greece-A	0.3944
Greece-D	0.1786	Saudi Aarbia	0.1780	Iran-Azeri	0.4072
Spain-Basques	0.1883	Southern Tunisia	0.1817	Moroccans-Jews	0.4130
Tunisia-Ghannouch	0.1917	Tunisia- Berbers-Met	0.1914	Lebanon-Yohmor	0.4397
Palestine	0.2118	Portugal	0.1995	Sri lanka	0.4477
Greece-A	0.2213	Morocco	0.2001	Libya-Jews	0.4537
Spain-Basques	0.2226	Libya	0.2022	Lebanon-B	0.4603
Ashkenazi-j	0.2264	France-Rennes	0.2024	Thailland	0.4689
Morocco-Jews	0.2450	Basques-A	0.2053	Lebanon-Armen	0.4896
Turkey-A	0.2564	France	0.2147	Bangladesh	0.4928
Spain-Berbers	0.2775	Tunisia-Berbers	0.2169	Turkey-A	0.5012
Lebanon-A	0.2817	Iran-Famoori	0.2170	Lebanon-A	0.5068
Guinea-Bubi	0.4143	Morocco-Chaouya	0.2170	Lebanon-KZ	0.5378
Senegal-Mandenka	0.4218	Yemen-sana’a	0.2278	Lebanon	0.5956
Mali-Fulani	0.4509	Pakistan-Sindh	0.2284	Senegal	0.6885
Mali- Mossi	0.5496	Tunisia-Ghannouchians	0.2292	Senegal-Mandenka	0.7174
Mali-Rimaibe	0.6432	Macedonia	0.2314	Iran-Jews	0.7476
		Basques-Arratia	0.2390	Lebanon-NS	0.7612
		Turkey	0.2394	Guinia-Bubi	0.7675
		Pakistan-Pathan	0.2414	Indonesia	0.9049
		India-Delhi	0.2465	Mali-Fulani	1.0674
		Yemen-Jews	0.2533	Mali-Mossi	1.3297
		Murcians	0.2616	Rimaibe	1.4181
		Greeks-B	0.2642		
		Iranians	0.2688		

* SGD: standard genetic distance.

## Data Availability

The data presented in this study can be found in the article/Appendix A; further inquiries can be directed to the corresponding author.

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
