# Peer review of "Analysis of the Origin of Emiratis as Inferred from a Family Study Based on HLA-A, -C, -B, -DRB1, and -DQB1 Genes"

_genes, 2023, doi:10.3390/genes14061159_

Round 1

Reviewer 1 Report

The authors reported the origin of Emirate and the relation between Emiratis and neighboring countries using appropriate genetic statistics methods and HLA family data.

Although there have been many anthropological analyses using HLA genes in this region, the present analysis is characterized by the large sample size and the analysis of five loci, but what differences are specific from the recent results reported by Alnaqbi A et al. (Ref 30, Sci Rep. 12,2022)?

Minor comments

1.       Please correct the “-“ between the HLA genes constituting the haplotype to “~“(e.g. Line27:DRB1*03-DQB1*02, Line142:B*08-DRB1*03, etc.)

2.       Please rewrite the alignment of HLA genes in the text in the order of HLA-A, -C, -B, -DRB1-DQB1, taking into account the chromosomal location of HLA genes and haplotype alignments. (e.g. Lien 117 and Table 1)

Author Response

Reviewer 1a: Although there have been many anthropological analyses using HLA genes in this region, the present analysis is characterized by the large sample size and the analysis of five loci, but what differences are specific from the recent results reported by Alnaqbi A et al. (Ref 30, Sci Rep. 12,2022)?

Response to reviewer 1a: There are no major differences between the two studies, however, our study was able to define a 5-loci HLA haplotype that is common in the Emirati population (HLA-A*26-C*07-B*08-DRB1*03-DQB*02). This was mainly related to the larger sample size that was used in our study. In addition, the publication by Alnaqbi et al. 2022 emphasized on 3-loci haplotype analysis. 

Reviewer 1b and 1c:

  1. Please correct the “-“ between the HLA genes constituting the haplotype to “~“(e.g. Line27:DRB1*03-DQB1*02, Line142:B*08-DRB1*03, etc.)
  2. Please rewrite the alignment of HLA genes in the text in the order of HLA-A, -C, -B, -DRB1-DQB1, taking into account the chromosomal location of HLA genes and haplotype alignments. (e.g. Lien 117 and Table 1)

Response to Reviewer 1b and 1c:

Both modifications were incorporated in the main text.

Reviewer 2 Report

It is a great paper, the hypothesis is original and the material is highly relevant.

The research has been conducted in high standards, the results are presented very well. All methods are described in great detail and are all repeatable and reproducible.

I do not have many comments. Maybe authors can discuss more about the potential disease risk profiles based on the most common HLA haplotypes in Emiratis. Can the authors draw any conclusions on the autoimmune and infection disease risks based on the HLA haplotype profiles?

Author Response

Reviewer 2: It is a great paper, the hypothesis is original and the material is highly relevant. The research has been conducted in high standards, the results are presented very well. All methods are described in great detail and are all repeatable and reproducible. I do not have many comments. Maybe authors can discuss more about the potential disease risk profiles based on the most common HLA haplotypes in Emiratis. Can the authors draw any conclusions on the autoimmune and infection disease risks based on the HLA haplotype profiles?

Response to reviewer 2: Since HLA typing was done mainly by low-to-intermediate resolution, we emphasized on well-established HLA disease associations with two autoimmune diseases, examples: (in the discussion part).

HLA-B51 is strongly and consistently associated with increased risk for Behçet’s Disease.

The HLA-DRB1*03-DQB1*02 haplotype was shown to be confer high risk for type 1 Diabetes.